# Predicting the Local Response of Esophageal Squamous Cell Carcinoma to Neoadjuvant Chemoradiotherapy by Radiomics with a Machine Learning Method Using ^18^F-FDG PET Images

**DOI:** 10.3390/diagnostics11061049

**Published:** 2021-06-07

**Authors:** Yuji Murakami, Daisuke Kawahara, Shigeyuki Tani, Katsumaro Kubo, Tsuyoshi Katsuta, Nobuki Imano, Yuki Takeuchi, Ikuno Nishibuchi, Akito Saito, Yasushi Nagata

**Affiliations:** 1Department of Radiation Oncology, Graduate School of Biomedical & Health Sciences, Hiroshima University, Hiroshima 734-8551, Japan; kubo1987@hiroshima-u.ac.jp (K.K.); tkatsuta@hiroshima-u.ac.jp (T.K.); imano@hiroshima-u.ac.jp (N.I.); ytake@hiroshima-u.ac.jp (Y.T.); ikuno@hiroshima-u.ac.jp (I.N.); akito@hiroshima-u.ac.jp (A.S.); nagat@hiroshima-u.ac.jp (Y.N.); 2School of Medicine, Hiroshima University, Hiroshima 734-8551, Japan; b160697@hiroshima-u.ac.jp

**Keywords:** esophageal cancer, squamous cell carcinoma, neoadjuvant chemoradiotherapy, pathological response, machine learning, radiomics

## Abstract

Background: This study aimed to propose a machine learning model to predict the local response of resectable locally advanced esophageal squamous cell carcinoma (LA-ESCC) treated by neoadjuvant chemoradiotherapy (NCRT) using pretreatment 18-fluorodeoxyglucose positron emission tomography (FDG PET) images. Methods: The local responses of 98 patients were categorized into two groups (complete response and noncomplete response). We performed a radiomics analysis using five segmentations created on FDG PET images, resulting in 4250 features per patient. To construct a machine learning model, we used the least absolute shrinkage and selection operator (LASSO) regression to extract radiomics features optimal for the prediction. Then, a prediction model was constructed by using a neural network classifier. The training model was evaluated with 5-fold cross-validation. Results: By the LASSO analysis of the training data, 22 radiomics features were extracted. In the testing data, the average accuracy, sensitivity, specificity, and area under the receiver operating characteristic curve score of the five prediction models were 89.6%, 92.7%, 89.5%, and 0.95, respectively. Conclusions: The proposed machine learning model using radiomics showed promising predictive accuracy of the local response of LA-ESCC treated by NCRT.

## 1. Introduction

Esophageal cancer is a malignant tumor that still has a poor prognosis. There are two distinct histological types, squamous cell carcinoma and adenocarcinoma, which predominate in East Asia and Western countries, respectively. For patients with resectable locally advanced esophageal squamous cell carcinoma (LA-ESCC), neoadjuvant treatment including neoadjuvant chemotherapy [1] or neoadjuvant chemoradiotherapy (NCRT) [2,3] has been shown to improve survival compared to surgery alone. Therefore, preoperative treatment followed by surgery has been the worldwide standard of care. However, the superiority of neoadjuvant chemotherapy or NCRT has not been determined at this time. On the other hand, definitive chemoradiotherapy (CRT) has been positioned as a treatment option for patients who wish to receive organ-preserving treatment or who are medically inoperable, because the results of definitive CRT to date have been somewhat inferior to those of surgery after neoadjuvant chemotherapy [4]. However, the Japan Clinical Oncology Group (JCOG) 0909 study recently reported that definitive CRT combined with salvage endoscopic resection or salvage surgery showed favorable overall survival and esophagectomy-free survival [5]. Thus, organ-preserving treatment will be a promising option for patients with resectable LA-ESCC in the future. However, the choice between surgery-based and organ-preserving definitive CRT-based treatment strategies is still difficult for patients.

In recent years, the introduction of radiomics and artificial intelligence (AI) into medicine has become a major topic of discussion. Radiomics is the study of systematically handling large amounts of imaging information in radiology [6]. The aim of radiomics is generally to extract quantitative, and ideally reproducible, information from diagnostic images, including complex patterns that are difficult to recognize or quantify by the human eye [7,8]. By combining radiomics and AI, attempts are being made to perform diagnostic imaging and to predict treatment outcomes [9,10,11,12,13].

18F-fluorodeoxyglucose positron emission tomography (FDG PET) image-derived parameters have been reported to be useful in predicting pathological response to NCRT and prognosis in patients with esophageal cancer [14,15,16]. In addition, the pathological complete response (pCR) rate after NCRT for resectable LA-ESCC was reported to be comparatively high [2,17]. However, since FDG PET imaging of the therapeutic effect of NCRT is performed immediately before surgery, it has not been possible to select a nonsurgical treatment. We, therefore, devised this study because we thought that if we could predict pathological response with high probability by analyzing medical images before NCRT using radiomics and AI technology, it would contribute to the treatment selection of resectable LA-ESCC patients. The purpose of this study was to propose a model to predict the local pathological response of resectable LA-ESCC patients after NCRT based on FDG PET images before NCRT by radiomics analysis and machine learning.

## 2. Materials and Methods

### 2.1. Patients

Eligibility for this study was based on the following criteria: a histologically confirmed thoracic esophageal or esophagogastric junction cancer; stage IB to IV disease without T4b lesions and distant metastasis other than supraclavicular lymph node metastasis (according to the 7th edition of the Union for International Cancer Control TNM Classification) diagnosed using endoscopy, computed tomography (CT), and FDG PET/CT; and receiving NCRT with cisplatin and 5-fluorouracil (5-FU) followed by surgery from 2003 to 2016. Histopathological response after NCRT was determined according to the 11th edition of the Japanese classification of esophageal cancer [18]. Grade 3 (no viable cancer cells are evident) in this classification corresponds to the pCR in this study. All patients provided written informed consent for treatment. The characteristics of the patients and their tumors are presented in Table 1.

### 2.2. FDG PET Image Acquisition

Patients fasted for at least 4 h before administration of 3.7 MBq/kg of FDG. The FDG PET/CT images were scanned using a Biograph mCT-64 (Siemens Healthcare, Erlangen, Germany). An unenhanced CT scan was performed with a 3 mm slice thickness. Both CT and PET scans proceeded under normal tidal breathing. An iterative algorithm with CT-derived attenuation correction was used for the reconstruction of FDG PET images.

### 2.3. Neoadjuvant Chemoradiotherapy Followed by Surgery

Treatment planning CT images were scanned under free breathing using a CT scanner (Light speed RT16; GE Healthcare, Little Chalfont, UK). The slice thickness was 2.5 mm. Three-dimensional radiotherapy treatment planning was performed. We delineated the gross tumor volumes for the primary tumor (GTVp), lymph node metastasis (GTVn), and both (GTVall). For the clinical target volume (CTV), CTVp and CTVn were defined as GTVp and GTVn plus a margin of 5 mm, respectively, and were adjusted according to the anatomical barrier. CTVsub was defined as elective nodal areas for subclinical lymph node metastasis. The elective nodal areas were determined according to primary tumor subsites as follows: supraclavicular to middle mediastinal nodal areas for upper thoracic tumors, upper mediastinal to perigastric nodal areas for middle and lower thoracic tumors, and lower mediastinal to celiac nodal areas for EGJ tumors. The volume including CTVp, CTVn, and CTVsub was defined as CTVall. The planning target volume (PTVall) was defined as the CTVall plus 8–12 mm margins. We used multiportal beams, if possible, to reduce the dose to the heart. Dose fractionation was 40 Gy in 20 fractions for PTVall. The concurrent chemotherapy regimen consisted of a combination of cisplatin (70 mg/m^2^ on days 1 and 29) and 5-FU (700 mg/m^2^/day on days 1–4 and 29–32). Surgery was performed 4 to 8 weeks after the completion of NCRT. The main surgical procedure was right transthoracic esophagectomy and two-field or three-field lymph node dissection. Patients with upper and middle thoracic esophageal lesions or lymph node metastasis in the upper mediastinum underwent cervical lymphadenectomy.

### 2.4. Process of the Radiomics Analysis

The process for generating a prediction model using a machine learning method with the radiomics feature is shown in Figure 1. The current study was designed as a Transparent Reporting of a Prediction Model for Individual Prognosis or Diagnosis (TRIPOD) type 2a [19].

#### 2.4.1. Preparation of Segmentation and Image Registration

We prepared five segmentations for radiomics analysis: GTVp, GTVp-2 mm, CTVp, CTVp-GTVp, and PTVall. “GTVp-2 mm” was defined as the inner tumor region of GTVp minus 2 mm of the outer edge. “CTVp-GTVp” was defined as the tumor peripheral region of CTVp minus GTVp. For each patient, these segmentations on the treatment planning CT were registered on the FDG PET/CT images using a deformable transformation field. The deformable image registration algorithm for the registration from the treatment planning CT images to the FDG PET/CT images consisted of two steps: a rigid image registration followed by a deformable image registration. By this process, each FDG PET voxel was mapped to a new position based on the transformations used in the CT-CT registration, resulting in a new FDG PET/CT dataset that was deformably registered with the treatment planning CT. All of the segmentations for radiomics analysis were performed by one or two radiation oncologists, including one expert radiation oncologist. Moreover, more than two researchers, including one expert radiation oncologist, checked and confirmed the performance of the deformable image registration and modified the segmentation if necessary.

#### 2.4.2. Radiomics Analysis

The pixel values of the FDG PET data were rescaled using the RescaleSlope and RescaleIntercept tags from the DICOM header as follows:Image Data = (Image Data) × RescaleSlope + RescaleIntercept + 1000(1)

The creation of radiomics features was performed using an open-source package in Python, Pyradiomics software [20]. The following features were created: morphology-based features (13 features), first order-based features (18 features), and texture analysis features, including Gray Level Co-occurrence Matrix (GLCM, 24 features), Gray Level Size Zone Matrix (GLSZM, 16 features), Gray Level Run Length Matrix (GLRLM, 16 features), Neighborhood Gray Tone Difference Matrix (NGTDM, 5 features), and Gray Level Dependence Matrix (GLDM, 14 features) (Table 2). Moreover, the FDG PET image data remained unchanged as the original, and these were preprocessed with a wavelet imaging filter. The wavelet filter has low-pass (L) and high-pass (H) filters. The decompositions were constructed in the x, y, and z directions. For example, “wavelet-HLL” was interpreted as a wavelet subband image resulting from directional filtering with a high-pass filter along the x-direction (H), a low-pass filter along the y-direction (L), and a low-pass filter along the z-direction (L). In the current study, eight wavelet subband images (wavelet-HLL, wavelet-LHL, wavelet-LHH, wavelet-LLH, wavelet-HLH, wavelet-HHH, wavelet-HHL, and wavelet-LLL) were created (Table 2). Each feature was computed separately with each of the above-mentioned preprocessing steps. From the above, a total of 850 features were created for each segmentation.

#### 2.4.3. Construction and Evaluation of Prediction Model

Among image features created by the radiomics technique, we selected the optimal features for machine learning using the least absolute shrinkage and selection operator (LASSO) logistic regression analysis with MATLAB code [21,22]. Furthermore, we used a machine learning method to construct predictive models, employing a neural network classifier with 10 hidden layers and rectified linear unit activation. The selected radiomics features were used as input values, and the information of “pCR” or “non-pCR” was used as output values. Here, 98 patients were randomly divided into a training group (72 patients; 54 for learning and 18 for validation) and a testing group (26 patients). To find the best predictive models, we used the 5-fold cross-validation method (Figure 2). The training–validation–testing processes were repeated five times for each patient group. The results of the prediction for the training and testing groups were evaluated in terms of accuracy, sensitivity, and specificity. Their predictive performance was evaluated using the area under the receiver operating characteristic (ROC) curve (AUC) score.

## 3. Results

A total of 4250 features were created from FDG PET images using the radiomics technique. In addition, 22 features for machine learning were selected using the LASSO analysis (Table 3). Twenty-one features were selected from the wavelet filtering features, and one feature was selected from the original image features. Regarding segmentations, eight features were selected from the GTVp, four from GTVp-2 mm, two from CTVp, three from CTVp-GTVp, and five from PTVall. All features were selected from the texture analysis of GLCM, GLRLM, GLSZM, and GLDM. The average values of each feature were compared in the pCR and non-pCR data. The average values of the Gray Level Variance, Gray Level Nonuniformity Normalized, Low Gray Level Run Emphasis, Low Gray Level Zone Emphasis, Gray Level Variance, Small Area Emphasis, Small Area High Gray Level Emphasis, Zone Percentage, Short Run High Gray Level Emphasis, and High Gray Level Run Emphasis in patients with pCR were smaller than those in non-pCR patients. On the other hand, the average values of the Small Area High Gray Level Emphasis, Correlation, MCC, High Gray Level Emphasis, and Large Dependence Low Gray Level Emphasis in pCR patients were higher than those in non-pCR patients.

Table 4 shows the performance of the NN models. There were five models generated in the 5-fold cross-validation step. The average accuracy of the five models was 95.2% (range, 92.7–98.2%) with the training group. The average accuracy, sensitivity, and specificity of the testing group were 89.6% (range, 87.0–95.7%), 92.7% (range, 84.6–100%), and 83.3% (range, 93.3–100%), respectively. Figure 3 shows the ROC curves of the predictive performance of five models for the testing group with 5-fold cross-validation. The AUC score was 0.93 for the 1st model, 0.94 for the 2nd model, 0.99 for the 3rd model, 0.92 for the 4th model, and 0.92 for the 5th model. The average and standard deviation of the AUC score with 5-fold cross-validation were 0.95 and 0.03, respectively.

## 4. Discussion

In this study, we constructed a model to predict the pathological response of primary tumors after NCRT for patients with resectable LA-ESCC using FDG PET image-based radiomics and machine learning analysis. Five-fold cross-validation analysis showed promising results with an average prediction accuracy of 89.5% (87.0–95.7%) and an average AUC score of 0.95 (0.92–0.99).

For patients with resectable LA-ESCC, neoadjuvant chemotherapy [1] or NCRT [2,3] followed by surgery is the standard treatment. At present, the superiority of NCRT over neoadjuvant chemotherapy remains unclear. To establish the superiority of NCRT over neoadjuvant chemotherapy, two randomized controlled trials were conducted. One was the Japanese three-arm trial, the JCOG1109 NeXT trial [23], for resectable LA-ESCC, and the other was the Irish Neo-AEGIS trial, ICORG10–14 [24], for resectable locally advanced adenocarcinoma. On the other hand, organ-preserving treatment strategies for resectable LA-ESCC have been investigated. Definitive CRT has been positioned as a treatment option for patients who wish to receive organ-preserving treatment or who are medically inoperable because the results of definitive CRT to date have been somewhat inferior to those of surgery after neoadjuvant chemotherapy [4]. However, the JCOG 0909 study recently reported that the 5-year overall survival rate of definitive CRT combined with salvage endoscopic resection or salvage surgery was 64.5%, and the esophagectomy-free survival rate was 54.9% [5]. Thus, organ-preserving treatment strategy will be a promising option for LA-ESCC patients in the future. However, the choice between surgery-based and organ-preserving, definitive CRT-based treatment strategies is still a very difficult issue for patients.

Regarding the pCR rate after NCRT in resectable LA-ESCC, van Hagen et al. reported that pCR rates in squamous cell carcinoma and adenocarcinoma were 49% and 23%, respectively [2]. Our previous study showed that the pCR rate after NCRT for patients with resectable LA-ESCC was 43% for primary tumors and 35% for both primary tumors and lymph node metastases [17]. In the patients enrolled in this study, NCRT resulted in pCR of the primary esophageal tumor in 44 of 98 patients (45%). Patients who achieved pCR after NCRT may have been cured by definitive CRT, and esophageal preservation may have been possible. If we can predict which patients with resectable LA-ESCC can achieve pCR by NCRT based on pretreatment medical information, it may contribute to the treatment selection of patients who wish to preserve their organs.

In recent years, the introduction of radiomics and AI into medicine has become a major topic. Regarding prediction of treatment outcomes, we reported that the neural network model using the radiomics features of tumor image was more accurate than the visual evaluation method using the image pattern information in predicting the local response of brain metastases to Gamma Knife radiosurgery [9]. Arshad et al., reported that PET image-based radiomics classifiers obtained prior to treatment were useful in predicting the prognosis of patients with non-small-cell lung cancer [10]. Peng et al., reported that PET/CT-based radiomics with deep learning for advanced nasopharyngeal cancer could serve as a prognostic tool and may act as an indicator for individualized induction chemotherapy [11]. Lv et al., reported that radiomics features extracted from the PET and CT components of baseline PET/CT images provide complementary prognostic information and improved outcome prediction for NPC patients compared with the use of clinical parameters alone [12]. Jiang et al., reported that the radiomics signature of PET/CT in gastric cancer patients was a powerful predictor of survivals and could predict which patients could benefit from chemotherapy [13].

FDG PET image-derived parameters have been reported to be useful in predicting the pathological response to NCRT and the prognosis of esophageal cancer patients [14]. We also reported that the rate of decrease in FDG uptake before and after NCRT in FDG PET images could be a prognostic factor [15,16]. However, it is necessary to predict the prognosis based on pretreatment information for patients who wish to undergo organ-preserving treatment to choose nonsurgical treatment. Regarding the study to evaluate the prediction of treatment response in esophageal cancer by radiomics using medical imaging, studies using CT and FDG PET images have been reported [25,26,27,28]. Hou et al. proposed a predictive model of tumor response using the pretreatment contrast-enhanced CT-based radiomics features of 49 patients with esophageal cancer. Their results showed that the classification accuracy using the neural network algorithm for testing cases was 0.917 and the AUC score was 0.800 [26]. Yang et al., developed three CT-based radiomics models for predicting pCR using data of 55 ESCC patients after NCRT. In their study, the AUC score in the testing group was 0.71–0.79 [27]. As for FDG PET imaging, recent studies reported that radiomics features of FDG PET images were better predictors of treatment response than the standard SUV method (SUVmax) [29,30,31]. Beukinga et al. constructed a model to predict the complete response to NCRT in esophageal cancer based on pretreatment clinical parameters and FDG PET/CT-derived radiomics features and reported that the predictive value of the constructed model was better than that of the SUVmax approach [31]. In their study, when the textual features of radiomics were introduced into the logistic regression analysis, the AUC score was 0.78 compared to 0.58 for the SUVmax model.

The current study constructed a predictive model for pathological findings after NCRT in patients with resectable LA-ESCC with FDG PET image-based radiomics and machine learning. For radiomics analysis, we constructed five segmentations of GTVp, GTVp-2 mm, CTVp, CTVp-GTVp, and PTVall.. Hao et al. developed the tumor shell as a radiomics feature that characterizes the tumor periphery and clarified the correlation between this feature and distant failure [32]. The ingenious things we have done in this study were constructing the inner tumor region (GTVp-2 mm) and tumor peripheral region (CTVp-GTVp) as segmentations. Of the 22 features extracted by LASSO analysis, 4 were related to the inner tumor region and 3 were related to the tumor peripheral region. In radiomics analysis, segmentation of not only the target itself but also the tumor peripheral regions and inner tumor regions may contribute to the creation of highly accurate models. Additionally, the LASSO analysis showed characteristic features for pCR and non-pCR cases. The pCR patients had radiomics features of small variance, high homogeneity, and high pixel values in GTVp and inner tumor regions. Moreover, radiomics features of fineness and coarseness by shell analysis can differentiate between pCR and non-pCR patients. In the CTVp and PTVall regions, the fineness and coarseness of the images were smaller in the pCR patients than in the non-pCR patients. This indicates that the images in and around the tumor of pCR patients were more homogeneous than those of non-pCR patients. Additionally, we used a 5-fold cross-validation method to evaluate the prediction model more accurately. As a result, our study showed promising results with a mean prediction accuracy of 89.5% (87.0–95.7%) and a mean AUC score of 0.95 (0.92–0.99). These results were more accurate than the results of previous studies of prediction models for esophageal cancer described earlier. The results also suggest that our prediction method using machine learning of radiomics features of pretreatment FDG PET may be more suitable for predicting pCR.

In radiomics, there are two main categories in imaging features: manually defined features and deep learning features. When compared with manually defined features, deep learning features are more specific to clinical outcomes and data [33]. In the current study, we used manually defined features. Although the results showed promising predictive value, the analysis was based on limited case data from a single institution. In the next step, we will need to validate the prediction model using more case data from multiple institutions. At that time, we plan to use various machine learning methods, including deep learning.

The current study has several limitations. In this prediction model, the pathologic response of only the primary esophageal tumor was considered, and not the pathologic response of lymph node metastases. It is unclear whether similar results will be obtained when the FDG PET imaging system is changed. This study was conducted at a single institution with a limited number of patients. To build a universal prediction model, we consider it necessary to examine a large number of cases in a multicenter setting. Moreover, the robustness of the performance of segmentation and deformable registration was not assessed in the current study. However, this report is significant in that it showed that a predictive model using radiomics and machine learning could significantly change the treatment choice for patients with resectable LA-ESCC in clinical practice in the future.

## 5. Conclusions

We constructed a model to predict the pathological response of primary tumors after NCRT for patients with resectable LA-ESCC using FDG PET image-based radiomics and machine learning, and the model showed promising prediction accuracy.

## Figures and Tables

**Figure 1 diagnostics-11-01049-f001:**
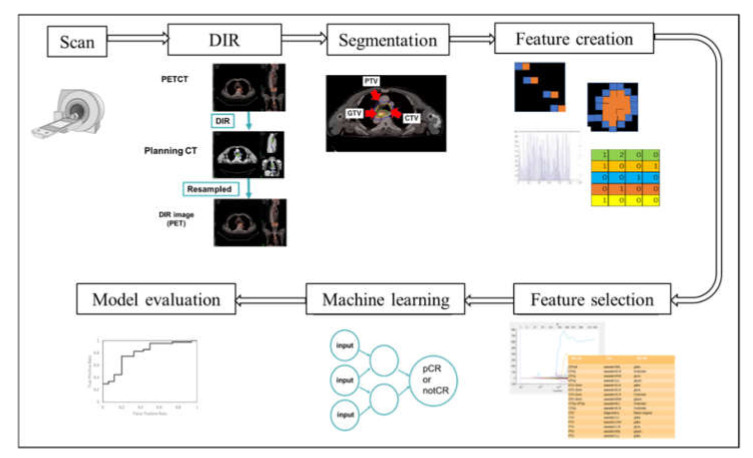
The process of the radiomics analysis and generating prediction model.

**Figure 2 diagnostics-11-01049-f002:**
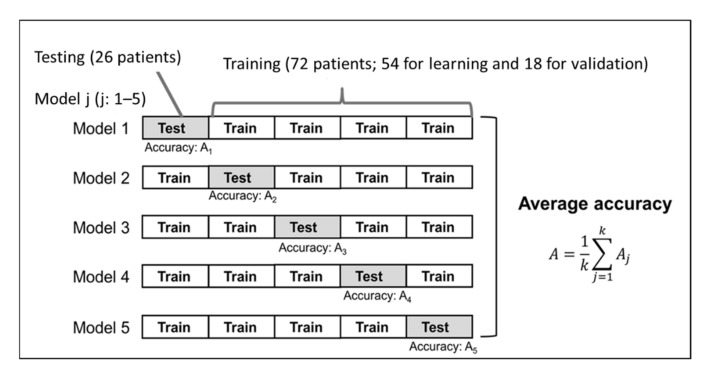
Generation and testing of the prediction model. The proposed neural network model with 5-fold cross-validation was built in the model training section.

**Figure 3 diagnostics-11-01049-f003:**
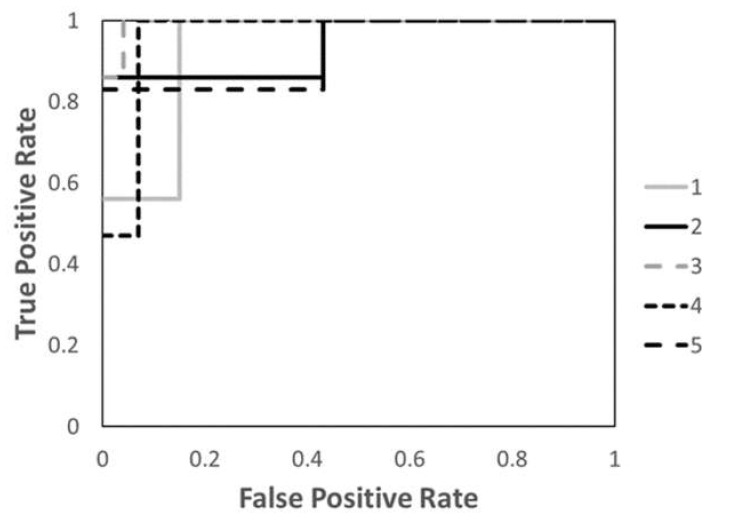
The area under the receiver operator characteristic curve (AUC) of five prediction models constructed by 5-fold cross-validation method. AUC scores for the models 1–5 were 0.93, 0.94, 0.99, 0.92, and 0.92, respectively.

**Table 1 diagnostics-11-01049-t001:** Characteristics of patients.

Characteristics		
Gender	Male/Female	83/15
Age (year)	Median (range)	66 (35–78)
Performance status	0/1/2	88/10/0
Tumor site	Upper/Middle/Lower-EGJ ^1^	22/46/30
T factor	1/2/3/4	2/16/79/1
N factor	0/1/2/3	19/52/25/2
Clinical stage	IB/II/III/IV	4/21/58/15
Local response	pCR ^2^/non-pCR	44/54

^1^ EGJ = esophagogastric junction, ^2^ pCR = pathological complete response.

**Table 2 diagnostics-11-01049-t002:** Feature type and associated features.

Feature Type	Morphology-Based	First Order-Based	Texture-Based				
Methods	Shape	Histogram	Gray Level Co-occurrence Matrix (GLCM)	Gray Level Size Zone Matrix (GLSZM)	Gray Level Run Length Matrix (GLRLM)	Neighborhood Gray Tone Difference Matrix (NGTDM)	Gray Level Dependence Matrix (GLDM)
Feature names	Maximum 3D diameter	Interquartile range	Joint average	Gray level variance	Short run low gray level emphasis	Coarseness	Gray level variance
	Maximum 2D diameter slice	Skewness	Sum average	Zone variance	Gray level variance	Complexity	High gray level emphasis
	Sphericity	Uniformity	Joint entropy	Gray level nonuniformity normalized	Low gray level run emphasis	Strength	Dependence entropy
	Minor axis	Median	Cluster shade	Size zone nonuniformity normalized	Gray level nonuniformity normalized	Contrast	Dependence nonuniformity
	Elongation	Energy	Maximum probability	Size zone nonuniformity	Run variance	Busyness	Gray level nonuniformity
	Surface volume ratio	Robust mean absolute deviation	Idmn	Gray level nonuniformity	Gray level nonuniformity		Small dependence emphasis
	Volume	Mean absolute deviation	Joint energy	Large area emphasis	Long run emphasis		Small dependence high gray level emphasis
	Major axis	Total energy	Contrast	Small area high gray level emphasis	Short run high gray level emphasis		Dependence nonuniformity normalized
	Surface area	Maximum	Difference entropy	Zone percentage	Run length nonuniformity		Large dependence emphasis
	Flatness	Root mean squared	Inverse variance	Large area low gray level emphasis	Short run emphasis		Large dependence low gray level emphasis
	Least axis	90 percentile	Difference variance	Large area high gray level emphasis	Long run high gray level emphasis		Dependence variance
	Maximum 2D diameter column	Minimum	Idn	High gray level zone emphasis	Run percentage		Large dependence high gray level emphasis
	Maximum 2D diameter row	Entropy	Idm	Small area emphasis	Long run low gray level emphasis		Small dependence low gray level emphasis
		Range	Correlation	Low gray level zone emphasis	Run entropy		Low gray level emphasis
		Variance	Autocorrelation	Zone entropy	High gray level run emphasis		
		10 percentile	Sum entropy	Small area low gray level emphasis	Run length nonuniformity normalized		
		Kurtosis	MCC				
		Mean	Sum squares				
			Cluster prominence				
			Imc2				
			Imc1				
			Difference average				
			Id				
			Cluster tendency				
Filtering	None	First-order statistic and texture of wavelet decomposition. Decomposition levels: LLL, LLH, LHL, LHH, HLL, HLH, HHL, HHH.					

**Table 3 diagnostics-11-01049-t003:** Selected features by LASSO ^1^ regression analysis.

Segmentation	Filter	Method	Feature Name	Relation of Average Value
GTVp ^2^	wavelet-LHH	GLRLM ^5^	Gray Level Variance	pCR ^9^ < non-pCR
GTVp	wavelet-LHH	GLRLM	Gray Level Nonuniformity Normalized	pCR < non-pCR
GTVp	wavelet-LLH	GLSZM	Gray Level Variance	pCR < non-pCR
GTVp	wavelet-LHH	GLRLM	Gray Level Variance	pCR < non-pCR
GTVp	wavelet-LHH	GLRLM	Low Gray Level Run Emphasis	pCR > non-pCR
GTVp	wavelet-LHH	GLRLM	Gray Level Nonuniformity Normalized	pCR > non-pCR
GTVp	wavelet-LLH	GLSZM ^6^	Gray Level Nonuniformity Normalized	pCR > non-pCR
GTVp	wavelet-HLH	GLSZM	Small Area High Gray Level Emphasis	pCR > non-pCR
GTVp-2 mm	original	GLSZM	Low Gray Level Zone Emphasis	pCR < non-pCR
GTVp-2 mm	wavelet-HLL	GLCM ^7^	Correlation	pCR > non-pCR
GTVp-2 mm	wavelet-HLL	GLCM	MCC	pCR > non-pCR
GTVp-2 mm	wavelet-HLH	GLSZM	Gray Level Variance	pCR < non-pCR
CTVp ^3^	wavelet-LHL	GLSZM	Gray Level Variance	pCR < non-pCR
CTVp	wavelet-LHH	GLDM ^8^	High Gray Level Emphasis	pCR > non-pCR
CTVp-GTVp	wavelet-LLH	GLSZM	Small Area Emphasis	pCR < non-pCR
CTVp-GTVp	wavelet-HLH	GLSZM	Small Area High Gray Level Emphasis	pCR < non-pCR
CTVp-GTVp	wavelet-HLH	GLSZM	Small Area Emphasis	pCR < non-pCR
PTVall ^4^	wavelet-LHL	GLDM	Large Dependence Low Gray Level Emphasis	pCR > non-pCR
PTVall	wavelet-HLH	GLSZM	Zone Percentage	pCR < non-pCR
PTVall	wavelet-HHH	GLRLM	Low Gray Level Run Emphasis	pCR > non-pCR
PTVall	wavelet-HHH	GLRLM	Short Run High Gray Level Emphasis	pCR < non-pCR
PTVall	wavelet-HHH	GLRLM	High Gray Level Run Emphasis	pCR < non-pCR

^1^ LASSO = the least absolute shrinkage and selection operator, ^2^ GTVp = gross tumor volume for primary tumor, ^3^ CTVp = clinical target volume for primary tumor, ^4^ PTVall = planning target volume for all targets, ^5^ GLRLM = Gray Level Run Length Matrix, ^6^ GLSZM = Gray Level Size Zone Matrix, ^7^ GLCM = Gray Level Co-occurrence Matrix, ^8^ GLDM = Gray Level Dependence Matrix, ^9^ pCR = pathological complete response.

**Table 4 diagnostics-11-01049-t004:** Model performance (%).

		Model 1	Model 2	Model 3	Model 4	Model 5	Average
Training	Accuracy	94.5	92.7	96.4	94.3	98.2	95.2
Testing	Accuracy	87	95.7	87	87	91.3	89.6
	Sensitivity	92.3	94.1	100	92.3	84.6	92.7
	Specificity	89	86	100	89	83.3	89.5

## Data Availability

The data presented in this study are available on request from the corresponding author.

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
