# Peer review of "Predicting the Local Response of Esophageal Squamous Cell Carcinoma to Neoadjuvant Chemoradiotherapy by Radiomics with a Machine Learning Method Using 18F-FDG PET Images"

_diagnostics, 2021, doi:10.3390/diagnostics11061049_

Round 1
Reviewer 1 Report
This is a retrospective monocentric study where the authors build a predictive model with internal validation using a 5-fold cross-validation technique. Two minor suggestions are needed:
- To specify which anatomopathological classification was used to consider the pathology response complete (TRG according to mandard? other?)
- - To specify level of validation of the predictive model by TRIPOD classification
Author Response
Dear reviewer 1,
Thank you very much for your valuable comments. We amended the paper according to your comments. Thanks to your comments, I think the paper is now better. Here is our response.
This is a retrospective monocentric study where the authors build a predictive model with internal validation using a 5-fold cross-validation technique. Two minor suggestions are needed:
- To specify which anatomopathological classification was used to consider the pathology response complete (TRG according to mandard? other?)
Answer: 
Thank you so much for pointing this out. We used the 11th edition of the Japanese classification of esophageal cancer to determine the histopathological response after NCRT. We added the description in the Materials and Methods section as follows.
“Histopathological response after NCRT was determined according to the 11th edition of the Japanese classification of esophageal cancer [18]. Grade 3 (no viable cancer cells are evident) in this classification corresponds to the pCR in this study.”
- To specify level of validation of the predictive model by TRIPOD classification
Answer:
Thank you for your valuable comment. We investigated a level of validation of the prediction model using TRIPOD classification. The current study used the dataset randomly split for validation. The TRIPOD type was 2a. We added the description in the material and method section as follows.
“The current study was designed as a Transparent Reporting of a prediction model for Individual Prognosis or Diagnosis (TRIPOD) type 2a [19].” 
And we added a reference article,
“Collins GS, Reitsma JB, Altman DG, Moons KG. Transparent reporting of a multivariable prediction model for individual prognosis or diagnosis (TRIPOD): the TRIPOD statement. Ann Intern Med. 2015, 162, 55-63, doi: 10.7326/M14-0697.”
Reviewer 2 Report
This study aims to propose a machine learning-based method to predict the local response of resectable locally advanced esophageal squamous cell carcinoma (LA-ESCC) treated by neoadjuvant chemoradiotherapy (NCRT) using pre-treatment 18-fluorodeoxyglucose positron emission tomography (FDG PET) images. The topic is interesting. My major comments are as follows.
- The representativeness of radiomics features may rely on the performance of segmentation and registration. Authors may need to provide more technique details of how they perform the segmentation and registration and how the performance of these two are (like inter- and intra- observer variation for segmentation and target registration error for deformable registration)?
- Is that possible that authors perform feature’s significance analysis for each kind of radiomics features?
- LASSO is used for feature selection. Recently, self-attention strategy in deep learning can also highlight the significant feature but in a supervised manner. Since the classification model used is neural network, how is the performance of replacing LASSO by self-attention?
- I suggest authors to provide details of the architecture of the used neural network, e.g., how many layers, what is the setting of each layer.
Author Response
Dear reviewer 2,
Thank you very much for your valuable comments. We amended the paper according to your comments. Thanks to your comments, I think the paper is now much better. Here is our response.
This study aims to propose a machine learning-based method to predict the local response of resectable locally advanced esophageal squamous cell carcinoma (LA-ESCC) treated by neoadjuvant chemoradiotherapy (NCRT) using pre-treatment 18-fluorodeoxyglucose positron emission tomography (FDG PET) images. The topic is interesting. My major comments are as follows.
- The representativeness of radiomics features may rely on the performance of segmentation and registration. Authors may need to provide more technique details of how they perform the segmentation and registration and how the performance of these two are (like inter-and intra- observer variation for segmentation and target registration error for deformable registration)?
Answer:
As the reviewer pointed out, the robustness of the performance of the segmentation and registration is an important issue for the radiomics analysis. The segmentation was performed by one or two radiation oncologists, including one expert radiation oncologist. And more than two researchers checked the performance of deformable image registration. There was no significant change in target position due to differences in upper limb position. So, we believe that the robustness of the radiomics features for inter-and intra-observer variation and deformable image registration was higher. However, we did not verify the performance of the registration data numerically.
We added the description about the method of target segmentation in the Material and Method section (2.4.1. Preparation of segmentation and image registration) as follows.
“All of the segmentations for radiomics analysis were performed by one or two radiation oncologists, including one expert radiation oncologist. Moreover, more than two researchers including one expert radiation oncologist checked and confirmed the performance of the deformable image registration and modified the segmentation if necessary.”
And we added the description in the limitation part as follows.
“Moreover, the robustness of the performance of segmentation and deformable registration was not assessed in the current study.”
- Is that possible that authors perform feature’s significance analysis for each kind of radiomics features?
Answer:
Thank you for your comment. For radiomics feature selection in this study, we used the LASSO regression analysis. So, we did not perform the feature’s significance analysis. There are other methods that can be performed the significant analysis such as multivariate analysis, Student t-test, Mann-Whitney U test, and so on. We will construct the prediction model with the multi-institutional data in the next step. Then we would like to perform the significant analysis for the feature selection and to use the other machine learning methods such as SVM, or a decision tree for the classification.
- LASSO is used for feature selection. Recently, self-attention strategy in deep learning can also highlight the significant feature but in a supervised manner. Since the classification model used is a neural network, how is the performance of replacing LASSO by self-attention?
Answer:
Thank you for your valuable comment. Radiomics was initially defined as the extraction of high-throughput features from images. Typically, there are two main categories in imaging features: manually defined features (including semantics and non-semantics) and deep learning features. We used manually defined features (non-semantics). And we quantified the extracted radiomics features and analyzed them as numerical data with a neural network. The self-attention strategy is the new topic for deep learning (for example, convolutional neural network). Deep learning analyses images directly. It is an interesting method, and it has the possibility to improve the accuracy of the prediction. We consider that the deep learning method is better used for predictive models with more patient data. So, we will use the deep learning method in the next step of the multi-institutional study.
We added the description about deep learning in the Discussion as follows.
“In radiomics, there are two main categories in imaging features: manually defined features and deep learning features. When compared with manually defined features, deep learning features are more specific to clinical outcomes and data [33]. In the current study, we used manually defined features. Although the results showed promising predictive value, the analysis was based on limited case data from a single institution. In the next step, we will need to validate the prediction model using more case data from multiple institutions. At that time, we plan to use various machine learning methods, including deep learning.”
And we add a reference,
“Liu, Z.; Wang, S.; Dong, D.; Wei, J.; Fang, C.; Zhou, X.; Sun, K.; Li, L.; Li, B.; Wang, M.; et al. The Applications of Radiomics in Precision Diagnosis and Treatment of Oncology: Opportunities and Challenges. Theranostics, 2019, 12, 1303-1322, doi: 10.7150/thno.30309.”
- I suggest authors to provide details of the architecture of the used neural network, e.g., how many layers, what is the setting of each layer.
Answer:
Thank you for your great suggestion. We used the neural network classifier with 10 hidden layers and rectified linear unit activation. The input data was used as the numerical data and the weight was optimized for each layer. We added the description in the material and method, as follows.
" Furthermore, we used a machine learning method to construct predictive models. The machine learning was used a neural network classifier with 10 hidden layers and rectified linear unit activation."
Round 2
Reviewer 2 Report
Authors answered all of my questions. No further amendment is needed.